# Effect of Heat Input on Microstructure and Mechanical Properties of Deposited Metal of E120C-K4 High Strength Steel Flux-Cored Wire

**DOI:** 10.3390/ma16083239

**Published:** 2023-04-20

**Authors:** Wen Wu, Tianli Zhang, Haoxin Chen, Jingjing Peng, Kaiqin Yang, Sanbao Lin, Peiyin Wen, Zhuoxin Li, Shanglei Yang, Sindo Kou

**Affiliations:** 1School of Materials Science and Engineering, Shanghai University of Engineering Science, Shanghai 201620, China; wuwenyyy@163.com (W.W.); chenhaoxin_95827@163.com (H.C.); peenny789@163.com (J.P.); ykq992711486@163.com (K.Y.);; 2State Key Laboratory of Advanced Welding and Joining, Harbin Institute of Technology, Harbin 150001, China; sblin@hit.edu.cn; 3State Tangshan Panasonic Industrial Machinery Co., Ltd., Tangshan 063020, China; 4College of Materials Science and Engineering, Beijing University of Technology, Beijing 100124, China; 5Department of Materials Science and Engineering, University of Wisconsin, Madison, WI 53706, USA

**Keywords:** heat input, high strength steel flux-cored wire, deposited metal, microstructure, inclusion, mechanical properties

## Abstract

The effect of different heat inputs of 1.45 kJ/mm, 1.78 kJ/mm and 2.31 kJ/mm on the microstructure and mechanical properties of deposited metals of the self-developed AWS A5.28 E120C-K4 high strength steel flux-cored wire was studied by optical microscope, scanning electron microscope and mechanical property test. With the increase in heat input, the results showed that the microstructure of deposited metals became coarse. Acicular ferrite increased at first and then decreased, granular bainite increased and degenerated upper bainite and martensite decreased slightly. Under the low heat input of 1.45 kJ/mm, the cooling rate was fast and the element diffusion was uneven, which caused composition segregation and easy to form large size inclusions SiO_2_-TiC-CeAlO_3_ with weak binding to the matrix. Under the middle heat input of 1.78 kJ/mm, the composite rare earth inclusions in dimples were mainly TiC-CeAlO_3_. The dimples were small and uniformly distributed, and the dimple fracture mainly depended on the wall-breaking connection between medium-sized dimples rather than an intermediate media. Under the high heat input of 2.31 kJ/mm, SiO_2_ was easy to adhere to high melting point Al_2_O_3_ oxides to form irregular composite inclusions. Such irregular inclusions do not need to absorb too much energy to form necking. Finally, the integrated effects of microstructure and inclusions resulted in the optimum mechanical properties of deposited metals with a heat input of 1.78 kJ/mm, which was a tensile strength of 793 MPa and an average impact toughness at −40 °C of 56 J.

## 1. Introduction

Low alloy high strength (HSLA) steel is widely used in ships, pipelines and other fields, so the requirements for matching welding consumables are also gradually improved. While ensuring the high strength of the deposited metals, to avoid brittle-fracture at low temperatures, the deposited metals should have adequate toughness. Therefore, to obtain an excellent balance between the strength and toughness of engineering materials at low temperatures, the choice of a suitable composition design and optimized welding parameters are very important [1,2,3,4,5]. Under the condition that the welding consumables are determined, the performance of deposited metal mainly depends on the welding process. The welding heat inputs have a large influence on microstructures, tensile strength and impact toughness. With the increase in the heat input, the increase in the contents of lath bainite and martensite-austenite (M-A) constituents lead to the increase in the strength. The coarsening of grain sizes and the formation of more M-A constituents at a higher input of 20.6 kJ/cm lead to the deterioration of the impact toughness and ultimate tensile strength of the weld metals [6]. Increasing the volume percentage of acicular ferrite (AF) in the deposited metal of HSLA steel can improve the mechanical properties, but lath bainite and coarse M-A constituents will decrease the mechanical properties [7,8,9]. Bhadeshia [10] demonstrated that the formation of AF was mainly determined by the austenite grain size, inclusions and the cooling rate. Lan [11] found that the medium cooling rate produced larger volume fractions of AF and that the size distributions of complex inclusions in the weld metal were very similar for all heat inputs, mainly because the short residence time did not make the inclusion grow rapidly. Song [5] found that low temperature toughness decreased with the increase in heat input, and that the impact fracture changed from ductile to cleavage fracture. In addition, the chemical compositions demonstrated a slight tendency to decrease with increasing heat input, while heat input had little effect on inclusions of deposited metal. Higher heat input led to further coarsened grains, reduced AF and granular bainite (GB) and increased polygonal ferrite, which mainly led to decreased toughness. Mainak [12] found that the volume fractions of AF in the weld metal increased with an increase in the thermal gradient and inclusion volume fraction. However, higher heat input, pulse frequency and thermal pulse frequency imparted a negative effect on the volume fractions of AF. Yang [13] demonstrated that the moderate welding heat input can help to avoid hot cracking. In addition, Dong [14] found that increasing the welding heat input could suppress the formation of martensite (M) and reduce the microhardness of heat affected zone (HAZ). However, the impact toughness of HAZ was not monotonously improved with the increase in welding heat input; the optimum comprehensive mechanical properties of HAZ in HSLA steel joints were achieved at a medium welding heat input (0.67 kJ/mm). The comprehensive heat input has a two-sided effect on the mechanical properties of the weld, which mainly controls the microstructure and mechanical properties by affecting the welding cooling rate and high temperature metallurgical reaction. The cooling rate mainly affects the formation of phase, and high temperature metallurgical reactions mainly affect the formation and distribution of inclusions [15].

Most research mainly focuses on the effect of the heat input and microstructure on the impact toughness of deposited metal. However, the presence of inclusion is generally thought to be a major factor that deteriorates the toughness of deposited metal [8,9,10,11,12,13,14,15,16,17,18]. Especially when the inclusion is modified by rare earth elements, because rare earth elements are sensitive to the change of welding thermal cycle temperature, the change of heat input will affect the reaction and existing state of rare earth elements and other alloy elements. The addition of rare earth Ce, Pr and Y have a good regulation effect on the deposited metal of high strength steel. The loss in the toughness of deposited metal is related to not only matrix microstructure, but also the formation of inclusions [19,20,21,22,23,24,25]. AWS A5.28 E120C-K4 high strength steel flux-cored wire has excellent resistance to wind, low spatter rate and all-position welding, high tensile strength and good toughness, even at low temperatures. Therefore, the self-developed AWS A5.28 E120C-K4 wire with the addition of 1% CeO_2_ was used in this paper. In this work, the effects of microstructure and inclusions morphology on the impact toughness of deposited metal were systematically investigated and discussed. It is expected to obtain the dialectical relationship among heat input, microstructure, inclusions and mechanical properties, so as to provide guidance for the subsequent optimization of deposited metal properties of high strength steel.

## 2. Materials and Methods

The welding parameters were adjusted for the single-pass welding test, and then a good process of three groups of heat input parameters was selected for the study of the deposited metals. The weld interpass temperature was controlled by infrared thermometer (DELIXI ELECTRIC, Wenzhou, China), weld beads were controlled in 6 layers of 12 passes, and after the completion of tensile tests and chemical composition tests, Charpy V-notched impact tests (HSLA steel belong to E steel, so specimens need to be placed in a mixture of −40 °C dry ice and absolute ethanol and thermal insulation for 20 min) were completed, the fracture specimen closest to the average impact energy was selected for fracture analysis at one end and the other end was prepared in accordance with GB/T13298-2015 standard for metallographic specimen. Specific experimental operations as follows:

The self-developed AWS A5.28 E120C-K4 wire (1.2 mm in diameter) with the addition of 1% CeO_2_ was used in the experiment. Deposited metal was prepared by mechanized welding according to the AWS standards A5.28 [26] and B4.0 [27]. The power source for welding was Cloos Pulse 450 (CARL CLOOS SCHWWISSTECHNIK GMBH, Haiger, Germany). The groove weld test assembly and pass arrangement for evaluation of mechanical properties are shown in Figure 1a. The welding parameters are shown in Table 1. The heat inputs were 1.45 kJ/mm, 1.78 kJ/mm and 2.31 kJ/mm, respectively. The corresponding deposited metals were designated as weld 1, weld 2 and weld 3, respectively. The base metal plates were ASTM A36 steels and their chemical composition are shown in Table 2. The groove faces and the contacting face of the backing steel of each weld surfacing to be 3 mm thick. After welding, the test plates were processed to make a round tensile specimen, as shown in Figure 1b, and the Charpy V-notched impact specimens, as shown in Figure 1c. The mechanical properties of deposited metal were determined by an AG-25TA (SHIMADZU, Kyoto, Japan) tensile testing machine and a TY10JBZ-300 (TIME GROUP Inc., Beijing, China) Charpy impact testing machine by GB/T 2560-2022 standards [28]. Five values of impact toughness at −40 °C were recorded, and the average value was taken after removing the maximal and minimal values. The chemical composition of the deposited metal was determined by Q4 optical emission spectrometer (BRUKE, Rheinstetten, Germany). One of the specimens was ground and polished to be the metallographic specimen etched by 4% nital. The quantitative statistical analysis was conducted for the microstructure of deposited metals with OLYCIA-M3 image analyzer (POOHER, Shanghai, China). The microstructure of the deposited metals and the morphology of impact fracture and inclusions were further examined by a TESCAN VEGA3 (TESCAN, Brno, Czech Republic) scanning electron microscope (SEM), and the composition of inclusions was analyzed by energy dispersive spectrometer (EDS). The deposited metals of the inclusion phase were detected by X-ray diffractometer (XRD) (BRUKE, Rheinstetten, Germany).

## 3. Results and Discussion

### 3.1. Mechanical Properties of Deposited Metal

The mechanical properties of deposited metal under different heat inputs are shown in Figure 2. When the heat input increased from 1.45 kJ/mm to 1.78 kJ/mm, the mechanical properties of the deposited metal increased, with the tensile strength increasing from 767 MPa to 793 MPa, the yield strength from 660 MPa to 700 MPa, and the average impact toughness at −40 °C increasing from 34 J to 56 J (increased by 65%). When the heat input was further increased to 2.31 kJ/mm, the mechanical properties of deposited metal decreased, and its value was similar to the low heat input value of 1.45 kJ/mm (764 MPa, 650 MPa and 34 J, respectively).

Figure 3 shows the SEM images of fracture surfaces of deposited metal under different heat inputs. The fracture surfaces are composed of dimple mixed with transgranular and intergranular cleavage fracture. As can be seen from Figure 3a,c,e, the cleavage fracture surface area first decreases and then increases with the increase in heat input; under the low heat input of 1.45 kJ/mm, the fracture surface has an obvious fluvial pattern and expands in one direction, which is relatively flat (Figure 3a), indicating that there is not much resistance to the expansion, and brittle fracture and grooves occur at the boundary of the small crystal plane [29]. Under the heat input of 1.78 kJ/mm, the cleavage surface was along different directions, there were obvious pits and the number of river patterns increased (Figure 3c). This is because the cleavage temperature increases with the increase in heat input, and a large amount of slip caused by cleavage leads to the accumulation of dislocations, resulting in an increase in the number of steps. Crack propagation requires more energy, and the cleavage surface tends to expand in areas with lower energy. When encountering obstacles, the direction of the cleavage surface changes repeatedly, so different cleavage directions are formed, resulting in varied dissociation facets, which is conducive to the improvement of toughness. Under the high heat input of 2.31 kJ/mm, the cleavage surface was similar to that under 1.78 kJ/mm, and there were many river patterns with different directions (Figure 3e). However, the relative area of each river pattern was larger than that of the river pattern in Figure 3c. With the increase of heat input, the burning loss of alloying elements reduces the dispersion strengthening effect of alloying elements on the matrix. In addition, the irregular high melting point inclusions that have not been modified by rare earth elements are easy to form crack sources at the tip. Coarsening grain and inclusions depress crack growth resistance of the materials strongly and result in the unstable propagation of cracks very easily [5,7]. Thereby, the strength of the deposited metal was reduced.

It can be seen from Figure 3b,d,f that with the increase in heat input, the percentage of spherical inclusions in the dimples gradually decreased, and the average pore diameter (φ_adv_) gradually increased. However, the maximum pore size (φ_max_) and the difference of dimple distribution size gradually decreased, and the dimple depth first increased and then decreased. Under the low heat input of 1.45 kJ/mm, due to the low heat input and short welding thermal cycle time, the rare earth Ce failed to adequately react with other elements to generate spherical composite inclusions in the metallurgical process, which is easy to form large composite inclusions in local areas. The dislocations bypass or cut through inclusions, so it is easy to form large holes. Deformation around the large holes is great, so it is easy to generate a series of shallow and small dimples. The fracture takes the small dimples as a bond to connect the medium-sized dimples to reduce the resistance to crack propagation, adversely affecting the toughness (Figure 3b). Under the heat input of 1.78 kJ/mm, more spherical composite inclusions of small size were generated due to metallurgical reactions, which promoted the formation of medium-sized dimples. Moreover, these dimples were uniformly distributed, and the deformation around the dimples was small, which was not enough to generate small dimples. Fracture can only occur after the fusion of these medium-sized dimples, which requires more energy. Therefore, good impact toughness can be obtained (Figure 3d). Under heat input of 2.31 kJ/mm, due to the increase in the peak temperature of the welding thermal cycle, it was easy to cause the burning loss of alloy elements, and the rare earth Ce in the deposited metal was less than 0.01%. Under this heat input, there were few spherical composite inclusions, and it was difficult to form large dimples. Additionally, with the slowdown of the cooling rate, the grain size increased. The energy fluctuations of the microstructure were small. Therefore, the barrier energy between the dimples was small and could fuse together and transform into worm-like dimples, which adversely affected the impact toughness (Figure 3f). Therefore, the impact toughness of deposited metal was the best under 1.78 kJ/mm heat input, and the impact toughness of deposited metal was poor under 1.45 kJ/mm and 2.31 kJ/mm heat input, which is consistent with the results in Figure 2.

Figure 4 shows the SEM images of fibrous zones on fracture surfaces and EDS spectrums of inclusions in deposited metal under different heat inputs. Figure 5 shows the schematic diagram of the ductile fracture mechanism under different heat inputs. It can be found that the composition, size and distribution of inclusions are different under different heat inputs, which directly affects the morphology and distribution of dimples. Rare earths exist mainly in the form of inclusions and rare earth–iron intermetallic compounds in deposited metal. Under the heat input of 1.45 kJ/mm, the dimple was mainly dependent on the medium size rare earth inclusion TiC-CeAlO_3_. In addition, the cooling rate was fast with low heat input, and the element diffusion was uneven, which caused composition segregation and easy to form large size inclusions SiO_2_-TiC-CeAlO_3_ with weak binding to the matrix, resulting in large pores. The surrounding deformation area generated shallow dimples because it absorbed enough stress (Figure 4b). The shallow and small dimples could be used as a connecting bridge between large holes and medium dimples to gradually connect them and generate fracture (Figure 5a). Under the heat input of 1.78 kJ/mm, the composite rare earth inclusions in dimples were mainly TiC-CeAlO_3_. The dimples were small and uniformly distributed, and the dimple fracture mainly depended on the wall-breaking connection between medium-sized dimples rather than an intermediate media, as shown in Figure 5b. Under the high heat input of 2.31 kJ/mm, a large number of alloy elements and rare earth elements were burned, and spherical rare earth composite inclusions were reduced. SiO_2_ easily adheres to high melting point Al_2_O_3_ oxides to form irregular composite inclusions. Such irregular inclusions do not need to absorb too much energy to form necking. After necking to a certain degree, shear fracture occurs to form small dimples, which are closely spaced and can be fuse together with each other. Worm-like dimples are formed, as shown in Figure 5c. It can be seen that Figure 4 confirms the phenomenon in Figure 3. The occurrence of dimple fracture may be due to the local strain concentration caused by the impact of the slip band on these second-phase particles, resulting in the nucleation of cavity. The small ductile fracture along these second-phase particles leads to ductile failure [30]. As shown in Figure 6, XRD were conducted on metallographic samples with different heat inputs, but the results were found to be consistent under different heat inputs; the measured inclusion phase included SiO_2_, Al_2_O_3_, TiC, CeAlO_3_, et al. This indicates that the heat input affects the mechanical properties of the deposited metals by changing the distribution of inclusions.

### 3.2. Microstructure of Deposited Metal under Different Heat Inputs

Figure 7 shows the microstructure of deposited metal under different heat inputs. It can be found that with the increase in heat input, the microstructure gradually coarsens. The AF increases at first and then decreases. The microstructure is mainly composed of AF, bainite and M, but also a small amount of cementite or M-A constituents. The percentage of each microstructure is shown in Figure 8. Under the low heat input of 1.45 kJ/mm, the grain size of the microstructure was smaller, and the percentage of M was higher. The crystallographic orientation of AF was poor, and the microstructure distribution was uneven. Under the heat input of 1.78 kJ/mm, the microstructure was mainly composed of AF and GB. Under the high heat input of 2.31 kJ/mm, AF and degenerate upper bainite (DUB) became obviously coarse, and some inclusions were dispersed at the boundary of DUB. On the other hand, heat input mainly affects weld microstructure by the control of high temperature metallurgical reactions and cooling rates (ν). According to the least square fitting and the formula [31], the relationship between t_8/5_, ν and heat input can be obtained by Formulas (1) and (2): E = 1.45 kJ/mm, t_8/5_ = 10 s, ν_1_ = 30 °C/s; E = 1.78 kJ/mm, t_8/5_ = 12 s, ν_2_ = 25 °C/s; E = 2.31 kJ/mm, t_8/5_ = 18 s, ν_3_ = 17 °C/s.
t8/5 = 4.79 × 10^−2^ × E^2^ − 1.0032 × E + 15.472(1)
ν_i_ = (800 °C − 500 °C)/t_8/5_(2)
t_8/5_: time for cooling the weld from 800 °C to 500 °C (s);E: heat input (kJ/mm);ν_i_: cooling rate (°C/s); i: 1, 2, 3.

It can be seen that with the increase in heat input, the ferrite and bainite become coarse because the high heat input slows down the welding cooling rate, the undercooling decreases, the phase transformation temperature increases and the free energy difference between the new and old phases decreases. The nucleation energy required for the formation of a new crystal nucleus increases and the nucleation rate decreases, so the ferrite and bainite lath formed gradually become coarse [32].

The alloy elements with low melting point will burn loss with the increase in heat input. Table 3 is the main chemical composition of deposited metal and mechanical properties; the remaining amount is Fe. It can be found that the contents of Si, Cr and Ni decrease. By substituting empirical Formulas (3) and (4), the transformation starting temperatures of bainite and M can be obtained: B_S_ (weld1) = 510.4 °C, B_S_ (weld2) = 515.5 °C, B_S_ (weld3) = 512.6 °C, M_S_ (weld1) = 434.5 °C, M_S_ (weld2) = 436.7 °C, M_S_ (weld3) = 437.0 °C [33]. It can be seen that the change of alloy elements on bainite transformation temperature is limited. Bainite transformation temperature is mainly affected by cooling rate. The faster the cooling rate is, the lower the bainite transformation temperature is. When bainite transformation occurs, the diffusion distance of C atom is shorter, and it is easy to accumulate on the interface of γ/α. However, the formation of cementite is habited because of the higher content of Si in deposited metal, leading to the formation of carbon-rich residual austenite at the boundary of bainite, so the final structure tends to form DUB [34]. The increase in heat input will lead to the burning loss of Si, which weakens the inhibition of cementite and reduces the percentage of DUB. The transformation temperature is generally higher of GB than the upper bainite. In this temperature range, with the increase in heat input, the bainite grows faster than the diffusion speed of C and the surrounding ferrite nucleates and grows; thus, C is gathered in a narrow area to form carbon-rich austenite. With the advance of the solidification process, M-A constituents are formed in the middle, creating the morphology of GB, so in a certain range of heat input, GB increases with the increase in heat input. On the other hand, because of the high undercooling, non-diffusion martensitic transformation will occur directly in some areas. The effect of heat input on AF is due to the fact that AF belongs to diffusive phase transformation, and the formation of AF is a process of nucleation and growth. With the increase in heat input, the high temperature residence time of liquid metal in the molten pool increases, which is beneficial to the nucleation and growth of rare earth composite inclusions in the liquid molten pool. The increase of the content of rare earth composite inclusions can form a coherent interface with α phase with minimum misalignment, which minimizes the surface energy of AF nucleation. Therefore, rare earth composite inclusions is beneficial to the formation of AF. However, on the other hand, the increase in heat input will lead to the burning loss of alloy elements and rare earth elements, thus reducing the nucleation rate of inclusions. Therefore, there is a heat input inflection point. When it is less than this heat input value, AF increases with the increase in heat input, but when it exceeds this inflection point, AF decreases instead.
B_S_ = 839 − (86χ_Mn_ + 23χ_Si_ + 67χ_Cr_ + 33χ_Ni_ + 75χ_Mo_) − 270 × [1 − e ^(−1.33χ^_C_^)^](3)
M_S_ = 565 − (31χ_Mn_ + 13χ_Si_ + 10χ_Cr_ + 18χ_Ni_ + 12χ_Mo_) – 600 × [1 − e ^(−0.96χ^_C_^)^](4)
B_S_: starting temperature for bainite phase transformation;M_S_: starting temperature for martensitic phase transformation;χ_i_: mass percent of each alloying element;i: Mn, Si, Cr, Ni, Mo and C.

### 3.3. Inclusions in Deposited Metal at Different Heat Inputs

With the increase in heat input, the residence time of high-temperature liquid of the molten pool increases, and the high-temperature non-metallic inclusions tend to nucleate and grow in the molten pool. However, when the heat input is too high, the alloying elements are burnt to different degrees, and the low-melting point inclusions also melt because of the high peak temperature, so the number of inclusions eventually increases first and then decreases. Observing the EDS spectrums in Figure 4, it can be seen that the inclusion phase mainly consists of Si, Mn, Ti, Al and other alloying elements and rare earth Ce, among which Si promotes the formation of SiO_2_; Mn promotes the formation of α-MnS; Ti can form TiO, Ti(O,N) or Ti(O,C); Al is easy to form Al_2_O_3_; and Ce is not easy to form a solid solution in austenite and tends to segregate at the grain boundary. Inclusions play a pinning role at the grain boundary, inhibiting the growth of austenite grain, as shown in Figure 9. In addition, the affinity of Ce to O and S is higher than that of other alloy elements, so it is easier to form spherical rare earth composite inclusions TiC-CeAlO_3_ and Ce_2_O_2_S at the grain boundary. The intragranular inclusions are mainly composed of ordinary oxide inclusions, which can promote the nucleation of AF and bainite in deposited metal under suitable conditions [35].

It can be found that heat input affects inclusions through the control of high-temperature metallurgical reactions and influences the microstructure through the control of the cooling rate, and ultimately achieves control of the mechanical properties of deposited metal so as to finally having an effect on the mechanical properties of deposited metal. The control mechanism is shown in Figure 10. With the increase in heat input, the peak temperature of the welding thermal cycle increases, and the high-temperature metallurgical reaction is more adequate. A large number of inclusions precipitate from the liquid, leading to different degrees of burning loss of Ce element. Therefore, the spherical rare earth inclusions first increase and then decrease, with some inclusions floating to the austenite grain boundaries. On the one hand, these inclusions play a role in pinning austenite grain boundaries, impeding the growth of austenite grains and playing a positive role in regulating the strength and toughness. On the other hand, they promote the formation of bainite ferrite at the grain boundary. Another part of high-temperature oxide inclusions dissociate into the austenite grain. As the nucleation particle of AF and bainite, under heat input of 2.31 kJ/mm, the undercooling degree was small and the energy difference between the old and new phases was low, manifesting the significant increase in the size of AF and bainite.

Therefore, under heat input of 2.31 kJ/mm, there were more GB as the toughening phase in austenite grains, and the AF content in the toughening phase was the same as that under the heat input of 1.45 kJ/mm. However, the existence of coarse microstructure cannot guarantee good toughness. As for strength, the DUB and M strengthening phases under the heat input of 1.45 kJ/mm were significantly higher than those under 2.31 kJ/mm, so the strength under 1.45 kJ/mm was higher. However, from the perspective of combined inclusions, low heat input is prone to form SiO_2_-TiC-CeAlO_3_ pore. As shown in the schematic diagram of the ductile fracture mechanism in Figure 5, the pores reduce the energy absorbed by dimple structure. Therefore, the impact toughness at 1.78 kJ/mm heat input was better than that at 1.45 kJ/mm. It can be concluded that the mechanical properties of deposited metal are the best at 1.78 kJ/mm, which is the result of the combined effect of the microstructure and rare earth composite inclusions.

## 4. Conclusions

(1)The heat input increased from 1.45 kJ/mm to 1.78 kJ/mm, the mechanical properties of deposited metal increased, with the tensile strength increasing from 767 MPa to 793 MPa, the yield strength from 660 MPa to 700 MPa, and the average impact toughness at −40 °C increasing from 34 J to 56 J (increased by 65%). When the heat input was further increased to 2.31 kJ/mm, the mechanical properties of deposited metal decreased, and its value was similar to the low heat input value of 1.45 kJ/mm (764 MPa, 650 MPa and 34 J, respectively). The best mechanical properties of deposited metal were obtained under the heat input of 1.78 kJ/mm.(2)When the heat input increased from 1.45 kJ/mm to 2.31 kJ/mm, the volume percentage of microstructure changed as follows: AF increased from 39% to 48%, and then gradually decreased to 37%; GB increased from 12% to 34%; DUB decreased from 40% to 26%; and M decreased from 9% to 3%. This is because that, the effect of heat input on the nucleation and growth process of AF has two sides; the increase in heat input promotes the metallurgical reaction and the dispersion distribution of rare earth composite inclusions, and the surface energy of AF nucleation decreases, which is beneficial to the formation of AF. On the other hand, the increase in heat input increases the size of AF, staggered aggregation and hinders the formation of new AF. The transformation temperature of GB is generally above the upper bainite transformation temperature. In this temperature range, with the increase in heat input, the growth rate of bainite is faster than that of C, and the surrounding ferrite is also nucleating and growing; thus, C is gathered in a narrow area to form carbon-rich austenite. With the advance of the solidification process, M-A constituents are formed in the middle, creating the morphology of GB, so in a certain range of heat input, GB increases with the increase in heat input. With the decrease in heat input, the cooling rate of DUB increases and the bainite transformation temperature decreases. When bainite transformation occurs, the shorter the diffusion distance of C atom is, so it is easy to gather on the interface of γ/α. However, because there is a higher content of Si in the deposited metal, it will inhibit the formation of cementite and lead to the formation of carbon-rich retained austenite at the boundary of B, so the final structure tends to form DUB. On the other hand, because of the high undercooling, non-diffusion martensitic transformation will occur directly in some areas with the decrease in the cooling rate.(3)With the increase in heat input, the spherical rare earth composite inclusions first increased and then decreased. Under the heat input of 1.45 kJ/mm, it was easy to gather and form large-size inclusions SiO_2_-TiC-CeAlO_3_, thus causing holes. Under the heat input of 1.78 kJ/mm, the inclusions mostly appeared in the form of TiC-CeAlO_3_. Under the heat input of 2.31 kJ/mm, SiO_2_ was easily attached to Al_2_O_3_ to form irregular composite inclusions. Heat input mainly controlled the inclusions through high temperature alloy reaction, and then regulated the microstructure volume ratio through the interaction of inclusions and the cooling rate, and finally indirectly affected the mechanical properties of deposited metal.

## Figures and Tables

**Figure 1 materials-16-03239-f001:**
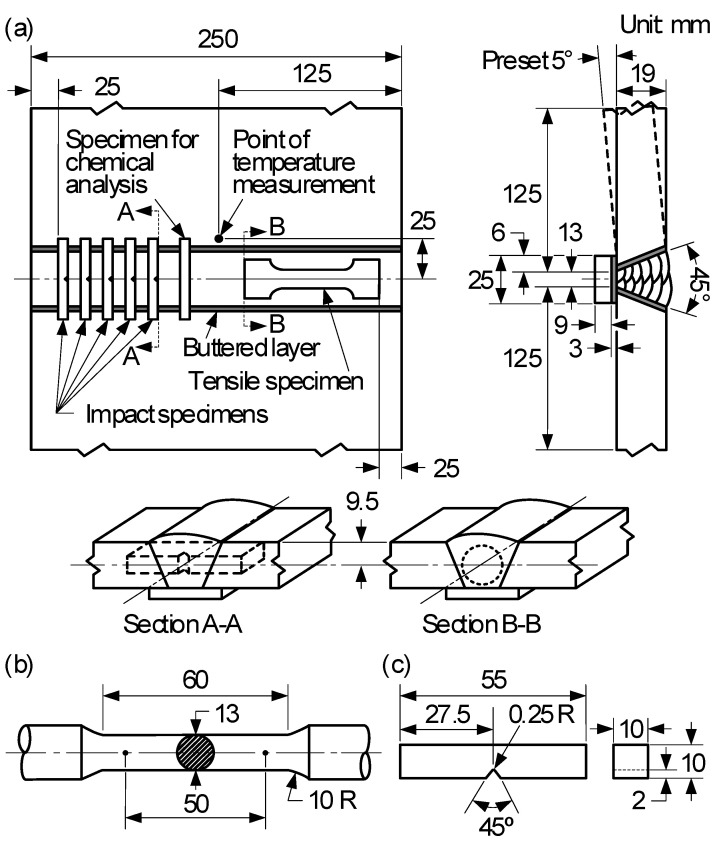
Groove weld test assembly and pass arrangement for evaluation of mechanical properties and dimensions of test specimens: (**a**) test plate showing locations and pass arrangement of test specimens; (**b**) dimensions of round tensile specimen; (**c**) dimensions of Charpy V-notch impact specimen [2].

**Figure 2 materials-16-03239-f002:**
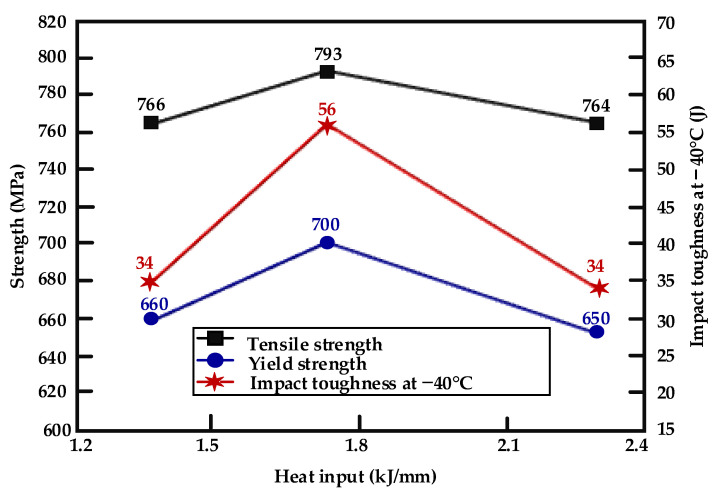
Results of mechanical properties of deposited metal.

**Figure 3 materials-16-03239-f003:**
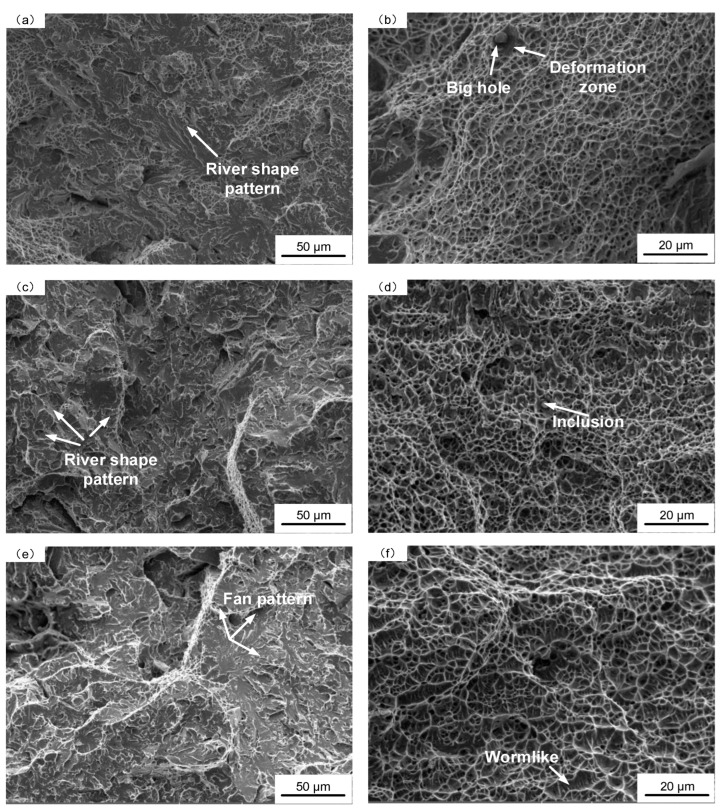
SEM images of fracture surfaces of deposited metal under different heat inputs: (**a**,**b**) 1.45 kJ/mm; (**c**,**d**) 1.78 kJ/mm; (**e**,**f**) 2.31 kJ/mm.

**Figure 4 materials-16-03239-f004:**
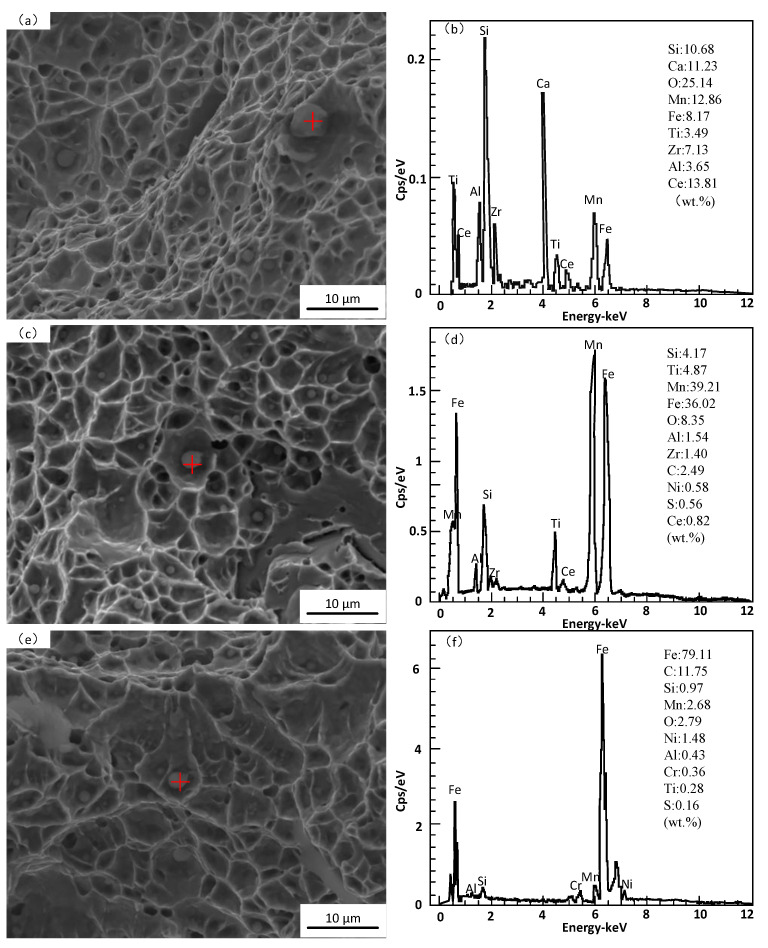
SEM images of fibrous zones on fracture surfaces and EDS spectrums of inclusions in deposited metal under different heat inputs: (**a**,**b**) 1.45 kJ/mm; (**c**,**d**) 1.78 kJ/mm; (**e**,**f**) 2.31 kJ/mm.

**Figure 5 materials-16-03239-f005:**
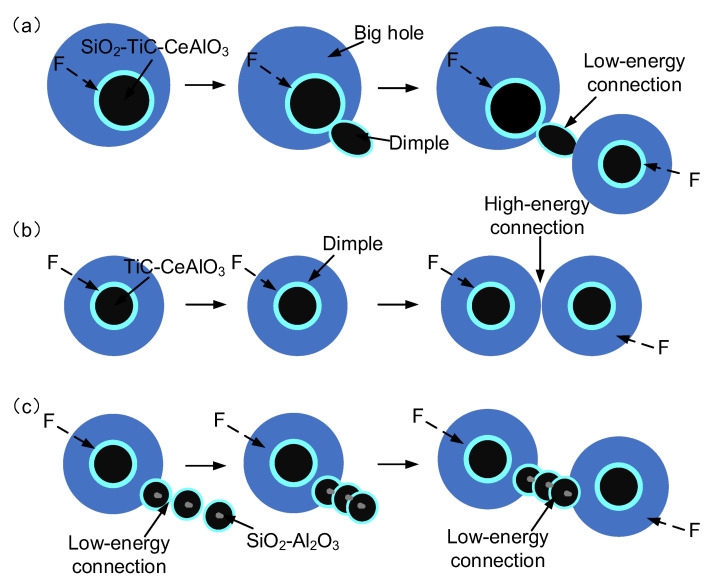
Schematic diagram of ductile fracture mechanism under different heat inputs: (**a**) 1.45 kJ/mm; (**b**) 1.78 kJ/mm; (**c**) 2.31 kJ/mm.

**Figure 6 materials-16-03239-f006:**
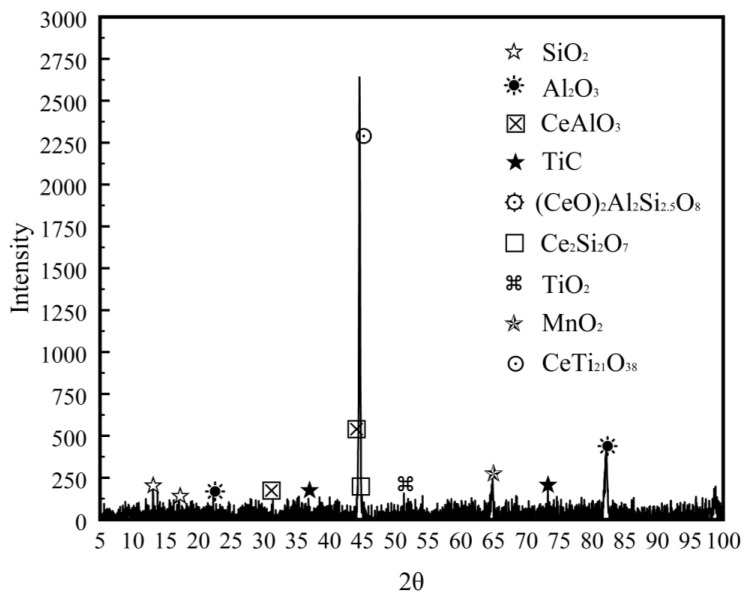
XRD image of deposited metals.

**Figure 7 materials-16-03239-f007:**
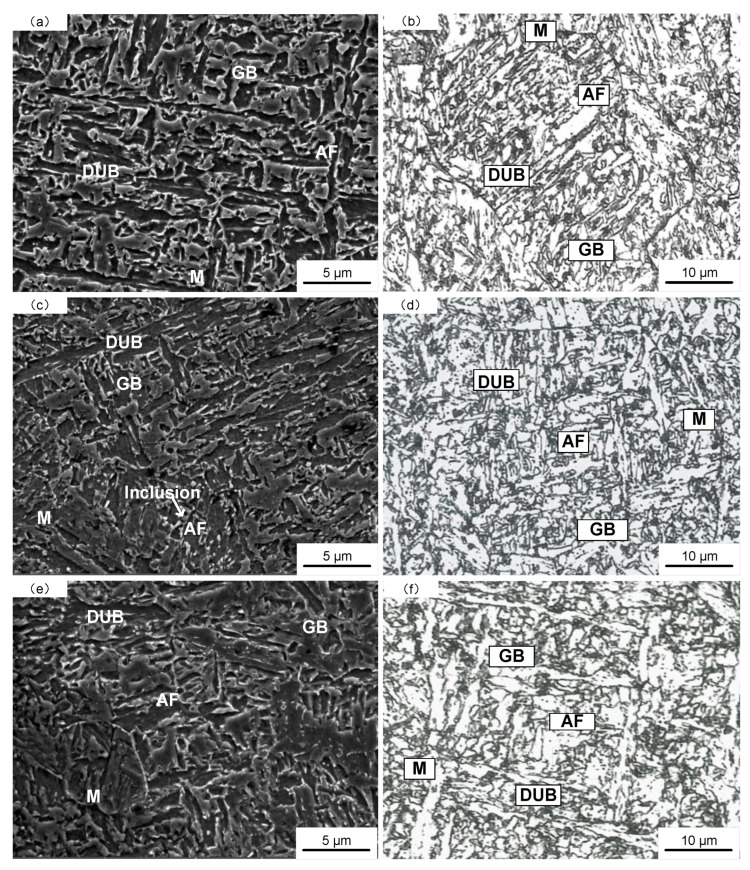
SEM images and optical micrographs of deposited metal under different heat inputs: (**a**,**b**) 1.45 kJ/mm; (**c**,**d**) 1.78 kJ/mm; (**e**,**f**) 2.31 kJ/mm.

**Figure 8 materials-16-03239-f008:**
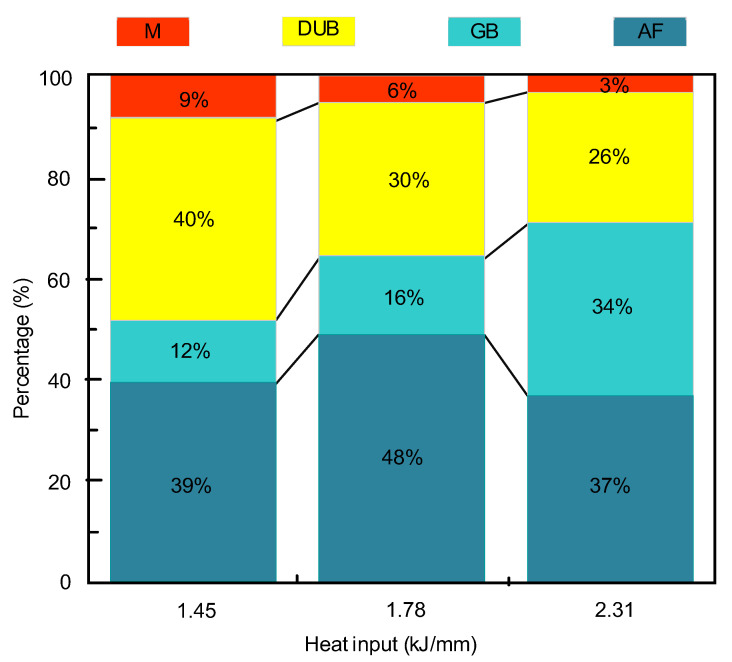
Quantitative statistical diagram of microstructure of deposited metal.

**Figure 9 materials-16-03239-f009:**
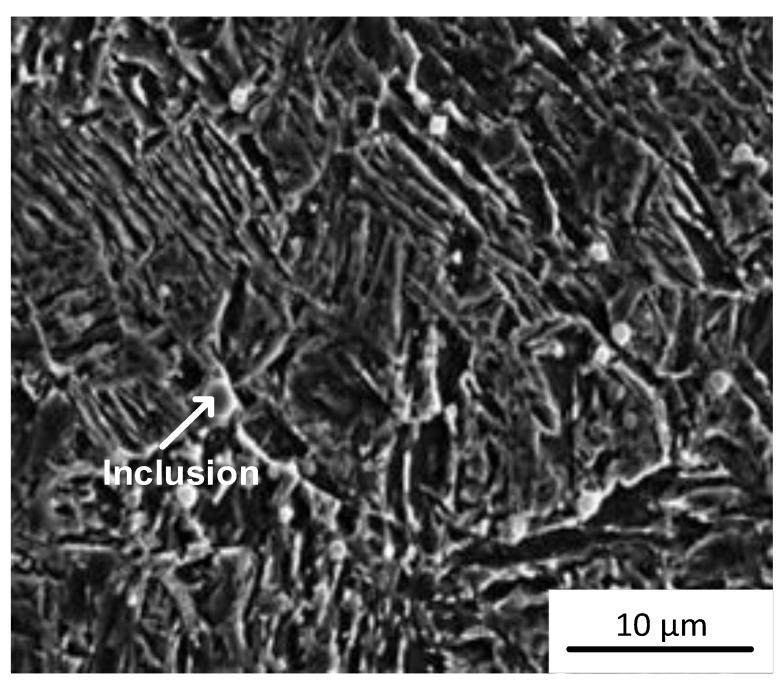
SEM image of rare earth inclusions pinning the austenite grain boundaries with the heat input 1.78 kJ/mm.

**Figure 10 materials-16-03239-f010:**
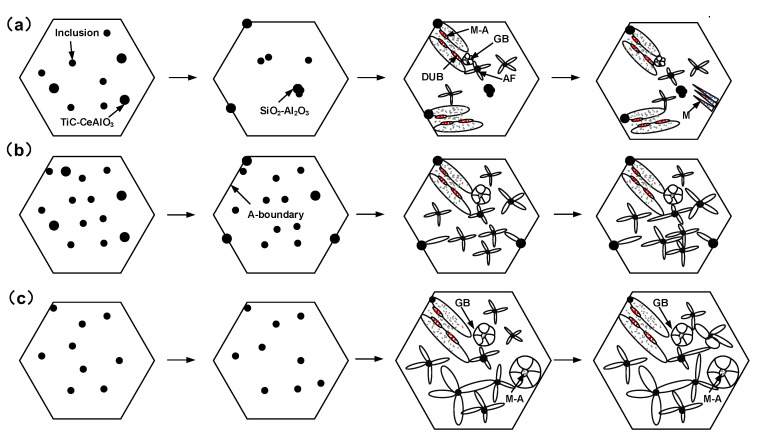
Schematic diagram of the relationship of inclusions, microstructure and mechanical properties under different heat inputs: (**a**) 1.45 kJ/mm; (**b**)1.78 kJ/mm; (**c**) 2.31 kJ/mm.

**Table 1 materials-16-03239-t001:** Welding parameters used in experiments.

No.	Voltage (V)	Current (A)	Wire Stick-Out (mm)	Welding Speed (cm/min)	Heat Input (kJ/mm)	Preheat/Interpass Temperature (°C)
Weld 1	22	230	15	21	1.45	150
Weld 2	25	250	15	21	1.78	150
Weld 3	30	270	15	21	2.31	150

**Table 2 materials-16-03239-t002:** Chemical composition of ASTM A36 steels.

Type	Chemical Composition (wt.%)					
	C	Si	Mn	P	S	Fe
ASTM A36	0.2	0.18	0.33	0.007	0.008	Bal.

**Table 3 materials-16-03239-t003:** Main chemical composition of deposited metal and mechanical properties (wt.%).

No.	Chemical Composition (wt.%)							Mechanical Property			
	C	Si	Mn	S	P	Cr	Ni	Mo	Tensile Strength	Akv (−40 °C)	
										Individual	Average
									(MPa)	(J)	
Weld 1	0.0255	0.49	1.94	0.007	0.01	0.391	2.13	0.60	766	33.6, 35.8, 33.2	34.2
Weld 2	0.0227	0.47	1.91	0.007	0.01	0.387	2.19	0.56	793	54.4, 57.7, 55.3	55.8
Weld 3	0.0396	0.33	1.97	0.007	0.01	0.388	2.07	0.60	764	38.3, 32.6, 31.1	34.0

## Data Availability

The data presented in this study are available on request from the corresponding author.

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
