# Peer review of "Effect of Heat Input on Microstructure and Mechanical Properties of Deposited Metal of E120C-K4 High Strength Steel Flux-Cored Wire"

_materials, 2023, doi:10.3390/ma16083239_

Round 1

Reviewer 1 Report

The reviewed manuscript entitled “Effect of heat input on microstructure and mechanical properties of deposited metal of E120C-K4 high strength steel flux-cored wire” investigates the effect of heat input on cooling rate, inclusion, microstructure and microstructure of high strength steel deposited metal. The article is made at a good scientific and technical level, and its practical significance is beyond doubt. In order to improve the readability and clarity of the manuscript, some major concerns need to be addressed before the paper is to be processed further:

1- According to which standard the tensile and impact tests were performed? Please add to the “Materials and Methods” section.

2- Figure 3: What is the scientific explanation for the deterioration of strength with an increase in Heat input to 2.31 kJ/mm? Discussion is lack of scientific explanation for the obtained results. Authors should attribute the results achieved to a clear scientific reason.

3- L167-175: The authors have built their interpretations on the formation of different compounds/phases based on a quantitative elemental analysis (EDS), While an acceptable evidence of the formation of these compounds/phases, such as X-ray diffraction test, is required as a proof.

4- Table 2: Just include the base matrix element, so the total will be 100wt%.

5- The English language used in the paper is to be revised and improved before the subsequent manuscript submission. Please, read the text carefully before the next submission of the paper.

Author Response

Thank you for your time and effort in processing the manuscript. We would like to express our sincere appreciations of your constructive comments concerning our article. Based on them, we have made careful modifications on the original draft.

The point-by-point responses are in the attachment.

Reviewer 2 Report

Comments to the Author

I have read the paper "Effect of heat input on microstructure and mechanical properties of deposited metal of E120C-K4 high strength steel flux- cored wire". The studies are really very interesting and produced good results, about the study effect of different heat inputs of 1.45 kJ/mm, 1.78 kJ/mm and 2.31 kJ/mm on the microstructure and mechanical properties of deposited metal of the self-developed AWS A5.28 E120C-K4 high strength steel flux-cored wire which is best suited for the “materials Journal”. The manuscript is structured nicely. However, I would propose that the author make the modest changes listed below.

-        I would suggest to modify some sentences that are not correct grammatically. However, the overall language of the paper is very simple and clear.

Abstract

Page 1, line 17, After every digit, kindly give the spacing like 1.45 kJ/mm, 1 % CeO2 etc. Kindly check the paper throughout and correct it.

Line 20, 25-26, 30-31, Kindly check the grammatical errors in the sentences. 

1.     Introduction

This part is well written, however, the description is very short, if possible, kindly elaborate it and add some latest references.

-kindly put spaces between the digits and units everywhere.

2.     Materials and Methods

Test methods and the description are provided nicely.

-        Page 2, line 73-74, Kindly check for grammatical errors and correct them. Somewhere present tense and some where past tense. I suggest kindly check throughout the paper and correct the sentences.

3.     Results and Discussion

This section is well discussed considering the test results.

3.1.  Mechanical properties of deposited metal

-        This subsection is explained clearly.

3.2.  Microstructure of deposited metal under different heat inputs

-        Page 8, lines 208–209. The authors mention E in kJ/mm and in Equation 1, referring to E in kJ/cm. If it is used in kJ/cm, then please provide the E values in kJ/cm in lines 208–209.

Also, I didn’t understand from where ν1, ν2 and ν3 values came here, kindly clarify it here, and if there is any equation, then kindly mention it here.

3.3 Inclusions in deposited metal at different heat inputs

- Well explained

4. Conclusions

Conclusions are supported in line with the presented results.

References

The latest reference (two) is from 2022, I would suggest adding some latest references if possible.

 The overall language of the paper is very simple and clear. Minor grammatical correction is required.

Author Response

(The authors gave the same response as above.)

Reviewer 3 Report

The review part of your article is very short. You write that previous authors get different results, therefore, your study is relevant. You should add details of what was done before you in this study direction, what problems the authors encountered, and how they solved them. Your study is not the first in changing the properties of steels and their welding/surfacing, so it is worth paying more attention to the review part.

At the beginning of the Materials and Methods, place a general plan of the experimental work.

Give tables with the chemical composition of the used wire and welded materials in the Introduction. Write how the chemical composition was determined: according to the reference book, or did you do the chemical analysis yourself?

For the equipment used, please indicate in brackets the manufacturer and the country of manufacture.

Expand the Materials and Methods section: describe in detail how the microstructure samples were cut, how many samples were taken, etc.

Break the section into paragraphs so that the study of microstructure and mechanical properties are separated (similarly for other types of studies). This will make the information easier for readers to understand.

You have a test at –40°C. From the Introduction and the first part of the Materials and Methods, it is not clear why you need them. This should be explained in the Materials and Methods section. Also, in the Introduction, please write about the need to study properties at negative temperatures in this study.

In Figure 2, the confidence level for the points should be indicated. In this figure, you have only 3 points: it is not clear what the nature of the curve will be with less or more heat input. Why in this case were limited to only three points? The nature of the curve between these three points is also unclear. Maybe the maximum lies between them, and not, as in the figure, in the region of the second point?

In the second conclusion, you write about the coarsening of the microstructure. Here it is better to say what quantitative characteristics of the microstructure have changed and how.

In Conclusion, you mainly give quantitative changes that occur during surfacing. It would be good to explain the course of these processes, which you have in the body of the article.

Author Response

(The authors gave the same response as above.)

Round 2

Reviewer 1 Report

The revision is satisfactory and the authors have provided amendments to all the suggested queries. Therefore I recommend this work for publication in Materials. 

Reviewer 3 Report

The authors did a good job of correcting the article. In my opinion, the article can be published.